

# Positivity, low twist dominance and CSDR for CFTs

Agnese Bissi[1][*] and Aninda Sinha[2][†]

**1** Department of Physics and Astronomy, Uppsala University,
Box 516, SE-751 20 Uppsala, Sweden
**2** Centre for High Energy Physics, Indian Institute of Science,
C.V. Raman Avenue, Bangalore 560012, India

[*] agnese.bissi@physics.uu.se , [†] asinha@iisc.ac.in

## Abstract

We consider a crossing symmetric dispersion relation (CSDR) for CFT four point correlation with identical scalar operators, which is manifestly symmetric under the cross-ratios $u, v$ interchange. This representation has several features in common with the CSDR for quantum field theories. It enables a study of the expansion of the correlation function around $u = v = 1/4$, which is used in the numerical conformal bootstrap program. We elucidate several remarkable features of the dispersive representation using the four point correlation function of $\Phi_{1,2}$ operators in 2d minimal models as a test-bed. When the dimension of the external scalar operator ($\Delta_\sigma$) is less than $\frac{1}{2}$, the CSDR gets contribution from only a single tower of global primary operators with the second tower being projected out. We find that there is a notion of low twist dominance (LTD) which, as a function of $\Delta_\sigma$, is maximized near the 2d Ising model as well as the non-unitary Yang-Lee model. The CSDR and LTD further explain positivity of the Taylor expansion coefficients of the correlation function around the crossing symmetric point and lead to universal predictions for specific ratios of these coefficients. These results carry over to the epsilon expansion in $4 - \epsilon$ dimensions. We also conduct a preliminary investigation of geometric function theory ideas, namely the Bieberbach-Rogosinski bounds.

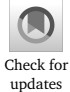

# 1 Introduction

Dispersion relations in the context of 2-2 scattering in quantum field theories have provided new insight about the structure of effective field theories [1–4]. Two sided bounds on the Taylor expansion coefficients of the scattering amplitudes follow from general considerations such as causality, unitarity and importantly, crossing symmetry. Crossing symmetric dispersion relations in [5] have led to establishing such bounds using elegant mathematical theorems arising from Geometric Function Theory [6–8].

In CFT, dispersive representations in $d \geq 2$ have been explored in [9–12]. In [9], the CFT analog of fixed-$t$ dispersion relation was considered. In this, the single discontinuity in the CFT s-channel plays a role. Crossing symmetry is not guaranteed and needs to be imposed by hand. In [10], sum rules have been derived using several equivalent dispersion relations. In this case, in the position space approach, one gets the so-called null constraints or odd-spin constraints on imposing crossing symmetry. This approach has yielded several interesting results numerically. Dispersive representations for CFT correlation functions have also been considered in Mellin space (see [13] for discussion). In [14], a systematic study was carried out establishing the nonperturbative existence of these amplitudes. A fixed-$t$ dispersion relation was written down and a study of the sum rules arising from this was initiated. In [15], the

crossing symmetric dispersion relation[1] for these amplitudes was considered and a derivation of the Polyakov bootstrap [17] was given.[2]

Motivated by the CSDR in QFT, one can ask if writing down a position space CSDR for CFT leads to new insights. We will show in this paper that the answer is yes. We will use 2d-CFT minimal models as a test bed [20–23]. This includes the diagonal unitary minimal models denoted by $M(m+1,m)$ as well as the Yang-Lee non-unitary model $M(5,2)$. In particular, following [20, 25], the external operator will be the $\Phi_{1,2}$ primary operator, which we will denote by $\sigma$. In the $\sigma \times \sigma$, OPE, the leading scalar operator is the $\Phi_{1,3}$ operator which we will denote by $\epsilon$ with twist $\tau_\epsilon$. In the $\sigma \times \sigma$ OPE, there are two towers of operators with twists $4k$ and $4k + \tau_\epsilon$, where $k = 0, 1, 2 \cdots$. We list out here some of the main advantages of the CSDR considered in this paper:

- The discontinuity involved is the single discontinuity as in [9]. However, we will exploit the better fall-off for minimal models which will lead to a different phase factor $(1 - \exp(-\pi i \tau))$, where $\tau$ is the twist of the exchanged operator in the OPE. This would mean that the $\tau = 4k$ tower gets projected out! This leads to better convergence using the CSDR. When the dimension of the external operator is more than $\frac{1}{2}$ or when we are in $d > 2$, the phase factor becomes $(1 - \exp(-\pi i (\tau - 2\Delta_\sigma)))$ as in [9], which projects out the generalized free field operators.

- We are interested in the Taylor expansion coefficients around the crossing symmetric point $u = v = 1/4$. This point played a role in the numerical CFT bootstrap [26]. In section 2, we will show a surprising positivity property of the Taylor expansion coefficients for both the unitary $M(m+1,m)$ minimal models as well as the non-unitary $M(5,2)$ Lee-Yang model. The CSDR will enable us to explain this feature using *low twist dominance* (LTD) (see fig.(3)), where in the calculation of the Taylor coefficients using the CSDR, the operator that dominates is simply the $\epsilon$ operator! In this sense, the expansion around $u = v = 1/4$ is like having an EFT expansion in quantum field theory with low spin dominance [27].

- We will derive an approximate formula for the Taylor coefficients (eq.6.6). Using the s-channel OPE, this leads to relations between such coefficients and the OPE coefficients of the two towers mentioned above. Moreover, we will be able to explain the positivity of the Taylor coefficients as well as universality of the ratios of specific coefficients that will be pointed out in the next section. These observations and predictions carry over to the epsilon expansion where the expression up to $O(\epsilon^2)$ for the correlator was worked out in [9].

- Finally, we will initiate a study of GFT bounds via the Bieberbach conjecture (de Branges theorem). This will enable us to distinguish the non-unitary Yang-Lee model from the unitary models (see fig(6)).

Another main advantage of considering CSDRs in the CFT context is that the minimal models provide an infinite family of crossing symmetric functions to study. In QFT, many assumptions have to be made about the analyticity and crossing symmetry. Some of these are more firmly established in CFT [28]. Furthermore, we do not have to be restricted to considering a weak coupling, as is often assumed in the QFT studies to avoid discussing logarithmic branch points. One of the main goals in the future will be to extend the ideas in this paper to the 3d Ising model and the $\epsilon$−expansion beyond $O(\epsilon^2)$ considered here. We do not envisage any conceptual difficulty in this.

---

[1]See [16] for an application.
[2]The Polyakov bootstrap in $d = 1$ was considered in [18, 19].

The paper is organized as follows. In section 2, we discuss the diagonal unitary Minimal models and point out several intriguing features. In section 3, we turn to the crossing symmetric dispersive representation. We begin with a general discussion and then focus on the Minimal models. However, note that all the formulas for the locality constraints and $\phi_{pq}$ apply more generally. In section 4, we consider the conformal block decomposition in the s-channel of these Minimal models, keeping crossing symmetry in mind. In section 5, we use the CSDR to demonstrate low twist dominance. In section 6, using the CSDR intuition, we explain the intriguing features pointed out in section 2. In section 7, we briefly consider Geometric Function Theory bounds for Minimal models. We end with future directions in section 8. The appendices contain useful supplementary calculations and details.

## 2 Positivity, clustering, universality$_\phi$ in Minimal models

We write the four point correlation function of identical scalar operators as

$$\langle \sigma(x_1)\sigma(x_2)\sigma(x_3)\sigma(x_4) \rangle = \frac{f(u,v)}{x_{12}^{2\Delta_\sigma} x_{34}^{2\Delta_\sigma}}. \tag{2.1}$$

Associativity of the OPE implies

$$f(u,v) = \left(\frac{u}{v}\right)^{\Delta_\sigma} f(v,u). \tag{2.2}$$

Often we use $u = z\bar{z}$, $v = (1-z)(1-\bar{z})$. Depending on the situation (Euclidean vs Lorentzian) $z, \bar{z}$ are either independent real variables (Lorentizan) or complex conjugate of one another (Euclidean). Motivated by early numerical bootstrap, we want to expand this around the crossing symmetric point $z = 1/2, \bar{z} = 1/2$ or $u = 1/4, v = 1/4$. For this reason we introduce $s_1 = u - 1/4, s_2 = v - 1/4$. Let us now discuss for concreteness the 2d-Ising model. Here $\sigma$ is the $\Phi_{1,2}$ operator. We have

$$f(u,v) = \frac{\sqrt{1 + \sqrt{u} + \sqrt{v}}}{\sqrt{2}\, v^{\frac{1}{8}}}, \tag{2.3}$$

with $\Delta_\sigma = \frac{1}{8}$. We define the crossing symmetric ($u, v$ interchange symmetric) object

$$F(u,v) = v^{\Delta_\sigma} f(u,v), \tag{2.4}$$

for example for the 2d Ising model we have

$$F(u,v) == \frac{1}{\sqrt{2}} \left( \sqrt{1 + \sqrt{s_1 + \frac{1}{4}} + \sqrt{s_2 + \frac{1}{4}}} \right). \tag{2.5}$$

More general correlators can be found using eq.(A.1 ). We will be interested in the Taylor expansion coefficients $\phi_{p,q}$ defined via

$$F(s_1, s_2) = \sum_{p,q} \phi_{p,q} x^p y^q, \tag{2.6}$$

where $x = s_1 + s_2, y = s_1 s_2$ are the two independent crossing-symmetric polynomials relevant for our purpose. For notational convenience, and since we will restrict to single digit $p, q$, we will use $\phi_{pq} \equiv \phi_{p,q}$. Following the parlance in the QFT literature, we will sometimes refer to the $\phi_{pq}$'s as Wilson coefficients.

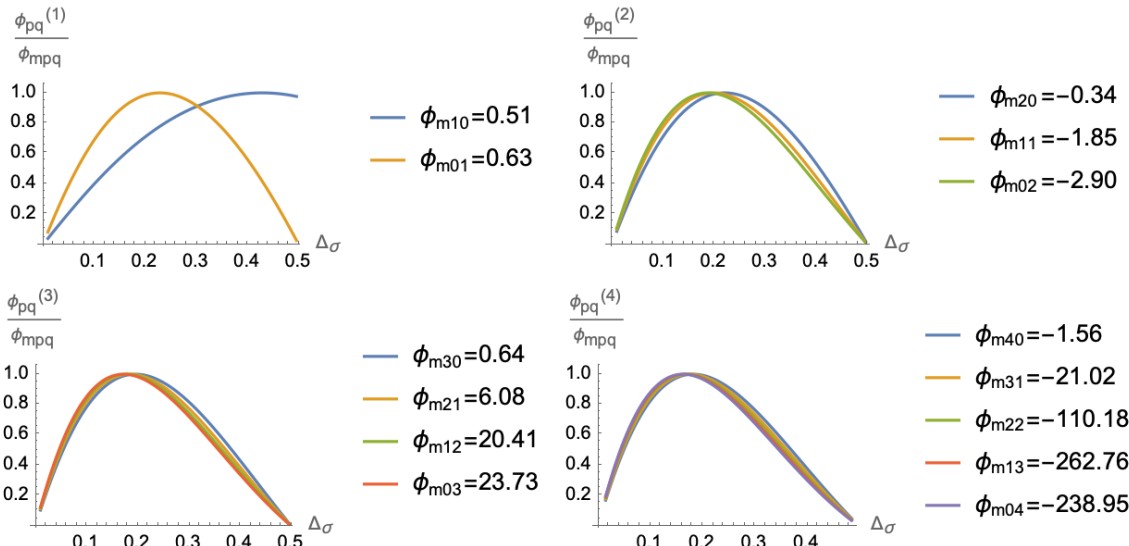

Figure 1: Plots of Wilson coefficients for minimal models described in the text. We have normalized appropriately as indicated in the legends for convenience.

We can consider the diagonal unitary minimal models in a similar manner. We will follow the notation in [20, 25]. We will denote by $\sigma$ the $\Phi_{1,2}$ operator and by $\epsilon$ the $\Phi_{1,3}$ operator.[3] Following [29], we can show that for minimal models we have in terms of the Virasoro primaries:

$$\sigma \times \sigma = \mathbb{1} + \epsilon. \tag{2.7}$$

This is consistent with our findings below that in terms of global primaries there are two infinite families of operators. The scaling dimensions for $\sigma, \epsilon$ are given by

$$\Delta_\sigma = \frac{1}{2} - \frac{3}{2(m+1)}, \quad \Delta_\epsilon = 2 - \frac{4}{m+1} = 2\frac{m-1}{m+1}, \tag{2.8}$$

The central charge is $c = 1 - 6/(m(m+1))$. The 2d Ising model corresponds to $m = 3$ while $m = 4$ is the tricritical Ising model. When $m \to \infty$ we have $\Delta_\sigma \to 1/2$, $\Delta_\epsilon \to 2$ and $c \to 1$. For later convenience, we tabulate some of the $\phi_{pq}$'s below while depicting the typical behaviour of the $\phi_{pq}$'s in fig.(1).[4]

| $m$ | $\Delta_\sigma$ | $\phi_{10}$ | $\phi_{01}$ | $\phi_{11}$ | $\phi_{20}$ | $\phi_{02}$ |
|---|---|---|---|---|---|---|
| 3 | $\frac{1}{8}$ | 0.250 | 0.500 | -1.625 | -0.281 | -2.625 |
| 4 | $\frac{1}{5}$ | 0.365 | 0.616 | -1.851 | -0.340 | -2.892 |
| 5 | $\frac{1}{4}$ | 0.424 | 0.620 | -1.762 | -0.337 | -2.688 |
| 6 | $\frac{2}{7}$ | 0.457 | 0.591 | -1.610 | -0.318 | -2.414 |
| 20 | $\frac{3}{7}$ | 0.514 | 0.262 | -0.587 | -0.134 | -0.816 |
| 50 | $\frac{8}{17}$ | 0.509 | 0.114 | -0.239 | -0.058 | -0.324 |
| 100 | $\frac{49}{101}$ | 0.505 | 0.058 | -0.120 | -0.029 | -0.161 |

We note the following observations:

- Let us define the level as $p + q$. Then for a given $p + q$, the plots indicate that $\phi_{pq}$ have the same signs. In other words, $(-1)^{p+q+1}\phi_{pq}$ are all positive.

---

[3]Note that $\sigma, \epsilon$ as denoted here are not the same as what is used in [29].

[4]$\phi_{10,max} \approx 0.5138$ at $\Delta_\sigma \approx 0.429$, while the maxima in $\phi_{pq}/\phi_{mpq}$ for $p + q \geq 2$ in the figures hover around $\Delta_\sigma \approx 0.16 - 0.19$.

- As $p + q$ increases, there is indication of "clustering", meaning all the normalized $\phi_{pq}$'s tend to lie on each other.

- There are further hidden patterns which are not evident. For instance $\phi_{02}/\phi_{40} \approx 2$, $\phi_{21}/\phi_{40} \approx -4$ etc for any $\Delta_\sigma$. This is illustrated later in fig.(5) and will be referred to as[5] "universality$_\phi$".

We have checked these features up to level 6 and they persist. In fact, as we will see, these persist even for non-unitary theories with $\Delta_\sigma > -0.5$. Our target in this paper is to find an explanation for all these features. This will demonstrate a concrete application (as well as show the advantage) of the CSDR. The bottom line is that these features are explained by *crossing symmetry* and *low twist dominance*.

The terminology *low twist dominance* (LTD) is motivated by low spin dominance, which is observed in EFTs [2, 27]. Since this may be unfamiliar, let us expand on what we mean by this in our context. First to use LTD, we have to specify what representation we are using. In our case, we will be using the crossing symmetric dispersive representation. What we mean by LTD is that the first few low twist operators contribute the most in the $\phi_{pq}$'s. As evidence, in appendix B, we estimate the contribution from the higher twist tail of the correlator for unitary theories [34]. We can make a drastic approximation and retain only the $\epsilon$-operator, i.e., only one operator. In the decomposition using s-channel blocks, we will not observe LTD, as we demonstrate below. As will become clear, we will assume LTD to explain the features stated in this section. In the appendix, will give a brief discussion as to how LTD could potentially be proved using the CSDR.

## 3 CSDR in position space

### 3.1 General dimensions

We are interested in CSDR in two variables. This was considered in [7] motivated by the old work [31] and revived in [5]. The problem being considered is the following. Let $g(u, v)$ be a function that satisfies crossing, namely $g(u, v) = g(v, u)$. Further we will choose the $u$-cut to be from $-\infty$ to 0. In [7], CSDRs were considered with different fall-offs. The one we will be interested in is the situation where $g(u, v)$ in the $|u| \to \infty$, fixed $v$, falls off faster than $|u|$. As we will discuss further below, this will enable us to consider both minimal models and the epsilon expansion. Writing $u = s_1 + \frac{1}{4}$ and $v = s_2 + \frac{1}{4}$, the CSDR becomes

$$g(s_1, s_2) = g(0, 0) + \frac{1}{2\pi i} \int_{-\infty}^{-\frac{1}{4}} \frac{ds_1'}{s_1'} A\left(s_1', \frac{as_1'}{s_1' - a}\right) H(s_1'; s_1, s_2), \qquad (3.1)$$

with the "absorptive part" defined as

$$A(s_1, s_2) = \underset{s_1}{\text{Disc}} \; g(s_1, s_2) = \lim_{\epsilon \to 0^+} [g(s_1 + i\epsilon, s_2) - g(s_1 - i\epsilon, s_2)], \qquad (3.2)$$

and where the kernel $H$ and the parameter $a$ are given by[6]

$$H(s_1'; s_1, s_2) = \frac{s_1}{s_1' - s_1} + \frac{s_2}{s_1' - s_2}, \quad a = \frac{s_1 s_2}{s_1 + s_2}. \qquad (3.3)$$

---

[5]We invented the terminology "universality$_\phi$" to prevent abusing the usual meaning of universality!

[6]The kernel is reminiscent of the tree level amplitude for scalar scattering $\psi_1 \psi_2 \to \psi_1 \psi_2$ mediated by a scalar $\phi$ of mass $s_1'$. The interacting lagrangian will be $s_1' \psi_1 \psi_2 \phi - (\psi_1 \psi_2)^2$. It is as if we are "averaging" over such theories.

Similar to the crossing symmetric variables used in [5,31], we can write

$$s_1 = a\left(1 - \frac{(1+\zeta)^2}{(1-\zeta)^2}\right), \quad s_2 = a\left(1 - \frac{(1-\zeta)^2}{(1+\zeta)^2}\right), \tag{3.4}$$

so that

$$x \equiv s_1 + s_2 = -16ak(\tilde{z}), \quad y \equiv s_1 s_2 = -16a^2 k(\tilde{z}), \tag{3.5}$$

where $\tilde{z} = \zeta^2$ and $k(\tilde{z})$ is the Koebe function in GFT having extremal properties:

$$k(\tilde{z}) = \frac{\tilde{z}}{(1-\tilde{z})^2} = \tilde{z} + \sum_{n=2}^{\infty} n\tilde{z}^n. \tag{3.6}$$

In the complex $\tilde{z}$ plane, the $u$-cut gets mapped to (portion of) the boundary of a unit disc. Depending on the range of $a$, the cut on the boundary either closes up or does not. For the range of $a$ we will consider, the cut does not close up allowing for analytic continuation from inside to outside. In terms of $\tilde{z}$, the kernel $H$ works out to be

$$H\left(\frac{a}{s_1'}, \tilde{z}\right) = 16\frac{a}{s_1'}\left(2\frac{a}{s_1'} - 1\right)\frac{\tilde{z}}{1 - 2\xi\tilde{z} + \tilde{z}^2} = -\alpha\partial_\alpha \ln[1 + 16\alpha(1-\alpha)k(\tilde{z})]\Big|_{\alpha=\frac{a}{s_1'}}, \tag{3.7}$$

with $\xi = 1 - 8\frac{a}{s_1'} + 8(\frac{a}{s_1'})^2$. The kernel can be identified as the generating function of the Chebyshev polynomials (in $\xi$) of the second kind and related to the Alexander polynomials of the torus $(2, 2n+1)$ knots with knot parameter $t$ defined via $2\xi = t + \frac{1}{t}$ as pointed out in [32].

As discussed below in section 3.3, for CFTs with identical scalars of dimension $\Delta_\sigma$, we have for fixed $s_2$, the large $s_1$ behaviour for the absorptive part to be $|s_1|^{2\Delta_\sigma}$. Therefore, for the CSDR to be applicable in the above form with

$$g(u, v) = F(u, v) \tag{3.8}$$

as in eq.(2.4), we will need $\Delta_\sigma < 1/2$. This will enable us to study the minimal models for $\Delta_\sigma < 1/2$. Using the notation in [9] we have the block decomposition

$$F(u, v) = v^{\Delta_\sigma} \sum_{\Delta,\ell} c_{\Delta,\ell} g_{\Delta,\ell}(u, v) \equiv \sum_{\Delta,\ell} c_{\Delta,\ell} F_{\Delta,\ell}(u, v). \tag{3.9}$$

In the small $u$ limit, we have $g_{\Delta,\ell}(u, v) \sim u^{\frac{\Delta-\ell}{2}}$. Then we find [9] that the discontinuity for $u < 0$ is given by

$$\mathrm{Disc}_{u<0} F_{\Delta,\ell}(u, v) = \left(1 - \exp\left(-2\pi i \frac{\Delta - \ell}{2}\right)\right) F_{\Delta,\ell}(u, v). \tag{3.10}$$

Thus even integer twists will get projected out. If we want to study the Wilson-Fisher fixed point e.g. the epsilon expansion, then we can define as in [9]

$$g(u, v) = \frac{F(u, v)}{(uv)^{\Delta_\sigma}} \equiv \sum_{\Delta,\ell} c_{\Delta,\ell} \widehat{F}_{\Delta,\ell}(u, v). \tag{3.11}$$

This makes the large $s_1$ limit of absorptive part to go like $|s_1|^{\Delta_\sigma}$. For the epsilon expansion, $\Delta_\sigma < 1$ so the CSDR is applicable. Here, we find [9] that the discontinuity for $u < 0$ is given by

$$\mathrm{Disc}_{u<0} \widehat{F}_{\Delta,\ell}(u, v) = \left(1 - \exp\left[-2\pi i\left(\frac{\Delta - \ell}{2} - \Delta_\sigma\right)\right]\right) F_{\Delta,\ell}(u, v). \tag{3.12}$$

Thus generalized free field (GFF) type operators will get projected out. Using the block decomposition in the absorptive part is justified in this case since $(\Delta - \ell)/2 - \Delta_\sigma > -1$ and there are no singularities introduced from the lower limit of the dispersive integral (see section 6.3). The differences between the two cases will be discussed further below. When $\Delta_\sigma > 1$, we will need to use the higher subtracted dispersion relation discussed in [7].

## 3.2 CSDR for minimal models

We will now focus on minimal models and for definiteness discuss the Ising model in eq.(2.4). The discontinuity across $z = 0$ is given by

$$\sqrt{2}\text{Disc}_z F(z,\bar{z}) = \sqrt{1 + \sqrt{z\bar{z}} + \sqrt{(1-z)(1-\bar{z})}} - \sqrt{1 - \sqrt{z\bar{z}} + \sqrt{(1-z)(1-\bar{z})}}. \qquad (3.13)$$

In the variables $s_1, s_2$ introduced above, $s_1 = 0, s_2 = 0$ is equivalent to expanding around $z = \frac{1}{2}, \bar{z} = \frac{1}{2}$. It is clear that for $|s_1| \to \infty$, fixed $s_2$, $F \to |s_1|^{2\Delta_\sigma} = \sqrt{|s_1|}$ and hence the CSDR applies. The CSDR for $F$ then reads

$$F(s_1, s_2) = \underbrace{F(0,0)}_{=1} + \frac{1}{2\pi i} \int_{-\infty}^{-\frac{1}{4}} \frac{ds_1'}{s_1'} A\left(s_1', \frac{as_1'}{s_1' - a}\right) H(s_1'; s_1, s_2), \qquad (3.14)$$

where $A(s_1, s_2)$ is given by

$$\sqrt{2}A(s_1, s_2) = \sqrt{1 + \sqrt{s_1 + \frac{1}{4}} + \sqrt{s_2 + \frac{1}{4}}} - \sqrt{1 - \sqrt{s_1 + \frac{1}{4}} + \sqrt{s_2 + \frac{1}{4}}}, \qquad (3.15)$$

The CSDR given in eq.(3.14) works when $s_1 > -\frac{1}{4}, s_2 > -\frac{1}{4}$ as we have verified numerically in a number of examples by comparing with the known answer.[7] For other minimal models, including the non-unitary Lee-Yang model, the analysis is similar.

The Ising model has two kinds of twists $\Delta - \ell = 4k, 4k + 1$ where $k \geq 0$ is an integer. All minimal models discussed below have the first tower of operators. The discontinuity then projects out these operators.[8]

## 3.3 Comments on convergence

We note here certain important points about the convergence of the conformal block expansion. The following statements are true for eq.(3.14).

- Inside the integral, notice that the CFT $z, \bar{z}$ are no longer complex conjugates of one another. In fact one can check that in the limit $s_1 \to -\infty$ with $s_2 = as_1/(s_1 - a)$, we have

$$\bar{z} = 1 + \frac{4a + 1}{4s_1} + O\left(\frac{1}{s_1^2}\right), \qquad (3.18)$$

$$z = s_1 - a - \frac{(2a + 1)^2}{4s_1} + O\left(\frac{1}{s_1^2}\right). \qquad (3.19)$$

---

[7]Using the known expression, we find that $A(s_1', as_1'/(s_1' - a)) > 0$ in the integration range for $-1/8 < a < 0$. Note that $\frac{a}{s_1'}(2\frac{a}{s_1'} - 1) < 0$ in the range of integration for $-1/8 < a < 0$ and has a definite sign. Thus, together with the $A$ factor in the kernel, this has a definite sign similar to the discussion of GFT methods for scattering amplitudes [6,7]. This is important to apply GFT techniques. In appendix E, we give an application of GFT.

[8]Although not directly relevant for the discussion of the dispersive integral, one can check that using eq.(3.12) we can reproduce the Taylor expansion of the discontinuity. For instance retaining the $4k + 1$ twist operators up to spin 6 and $k_{max} = 4$ we get

$$0.292893 + 0.603554s_1 - 0.103553s_2 - 0.546419s_1^2 - 0.239295s_1s_2 + 0.160674s_2^2 + \cdots, \qquad (3.16)$$

from the block expansion whereas the expected answer is

$$0.292893 + 0.603553s_1 - 0.103553s_2 - 0.546415s_1^2 - 0.239277s_1s_2 + 0.160692s_2^2 + \cdots \qquad (3.17)$$

Increasing the spins to 10, the minor discrepancies go away.

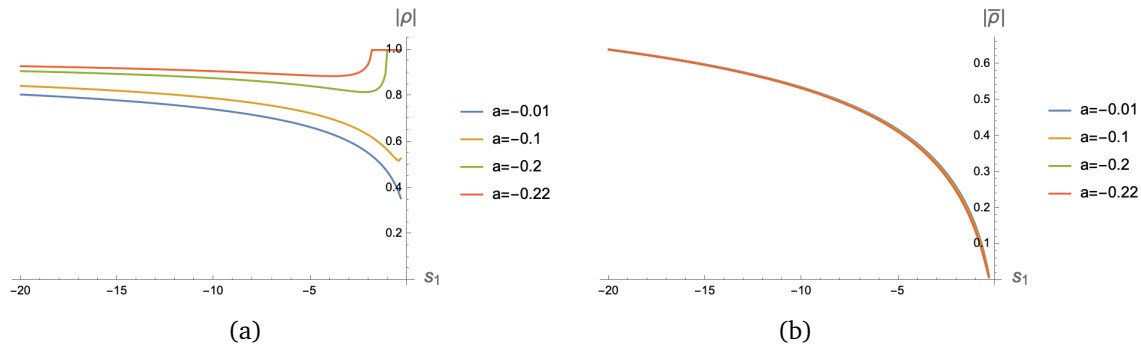

Figure 2: (a) $|\rho|$ vs $s_1$. (b) $|\bar{\rho}|$ vs $s_1$.

Thus $\bar{z} \to 1_-$ while $z \to -\infty$ (or $u \to -\infty$, $v \to a + \frac{1}{4}$). The discussion of convergence then follows that of [33]. In the variables

$$\rho = \frac{z}{(1 + \sqrt{1-z})^2}, \quad \bar{\rho} = \frac{\bar{z}}{(1 + \sqrt{1-\bar{z}})^2}, \qquad (3.20)$$

in the integration domain we find that $|\rho| \le 1, |\bar{\rho}| \le 1$ for any real $a$ but $|\rho| < 1, |\bar{\rho}| < 1$ for $a > -\frac{1}{8}$ as the plots below illustrate. In fig.2 we find that $|\rho| = 1$ for a range of $s_1$ when $a \le -1/8$. Since we wish to work with the $\phi(x_1) \times \phi(x_2)$ OPE channel in the dispersive integral, we will only focus on $-1/8 < a < 0$. This range ensures that the $\xi$ variable in the dispersive representation satisfies $|\xi| < 1$ so that there are no singularities inside the unit disc $\tilde{z} \le 1$. This will be needed to discuss the GFT bounds on the Taylor expansion coefficients.[9] General considerations [33] lead to $|f(\rho, \bar{\rho})| < (1 - r)^{-4\Delta_\sigma}$ where $r = max(|\rho|, |\bar{\rho}|)$. In the dispersive integral, this will translate to $|f| < |s_1|^{2\Delta_\sigma}$ for large $|s_1|$.

- With this fall-off, for the minimal models the integrand at large negative $s_1'$ behaves like $O(\frac{1}{|s_1'|^{2-2\Delta_\sigma}})$. This means that the integral converges provided $\Delta_\sigma < 1/2$. If instead of eq.(3.13), we use $F(u,v)/(uv)^{\Delta_\sigma}$, then the integrand in the dispersive integral would behave like $O(\frac{1}{|s_1'|^{2-\Delta_\sigma}})$ which would improve the range of $\Delta_\sigma$ to $\Delta_\sigma < 1$. Now, we would get the projection factor $(1 - \exp(-2\pi i(\tau/2 - \Delta_\sigma)))$ which would project out GFF operators [9]. This would be relevant for Wilson-Fisher fixed points. Note that arbitrary insertions of $(uv)^{\#}$ in defining $g(u,v)$ in the dispersive representation is problematic since the lower limit of the dispersive integral ($u = 0$) could blow up. For the minimal models and for the epsilon expansion, we do not encounter this problem.

## 3.4 Locality constraints

As in the QFT case [5], the penalty for keeping the $s_1 \leftrightarrow s_2$ symmetry manifest is the loss of manifest "locality". This means that while expanding the known answer around $s_1 = 0, s_2 = 0$ we get only polynomials in $x, y$ defined in eq.(3.5), the combination $AH$ in the dispersion relation will have negative powers of $x$. The way to see this is to note that $H$ on its own is "local" in the above sense. However, $A(s_1', as_1'/(s_1' - a))$ can be Taylor expanded around $a = 0$ to yield arbitrary powers of $a^n$. Since $a = y/x$, this would potentially need to arbitrarily negative powers of $x$. The cancellation of these negative powers are what were dubbed "locality" constraints in [5, 15] and we will continue to use the same terminology. Negative powers of

---

[9]A more general discussion on the bounds allowing for singularities inside the unit disc is possible following [7, 35] but we will not do that in this work.

$x$ would correspond to poles in the correlation function when $u + v = 1/2$ which should be absent in CFTs. We have checked with the known expressions that indeed all such potentially negative powers of $x$ cancel out. What is nontrivial, however, is to examine their cancellation at the level of the block expansion.

Each block will lead to such negative powers of $x$ and will be "non-local". It is only when the operators are summed over that these powers will cancel. In the fixed-$t$ dispersion, one imposes the constraints arising from crossing symmetry and finds the so-called "null constraints" [4]. The locality constraints are their counterpart. The challenge for us now is to efficiently extract, block-wise, these negative powers of $x$. This we turn to next; the discussion below is general and follows from eq.(3.1). Noting that inside the dispersive integral $v = \frac{as_1}{s_1 - a} + \frac{1}{4}$, we can write (anticipating that $s_1$ will be integrated over; we drop the prime for convenience)

$$A = \sum_{m=0}^{\infty} f_m(s_1)\left(\frac{a}{s_1}\right)^m. \tag{3.21}$$

Now using eq.(3.7) we can write

$$H\left(\frac{a}{s_1}, \tilde{z}\right) = \sum_{n=1}^{\infty} k(\tilde{z})^n c_n\left(\frac{a}{s_1}\right), \quad c_n(\alpha) = 16\alpha(2\alpha-1)(-16\alpha(1-\alpha))^{n-1} \equiv \sum_{k=0}^{2n} \chi_{n,k} \alpha^k. \tag{3.22}$$

When we consider the dispersive integral, the statement of locality is simply the following: For $k(\tilde{z})^n$, after the dispersive integral, the maximum power of $a$ should be $a^{2n}$. It can be checked that this translates into the condition

$$\int_{-\infty}^{-\frac{1}{4}} \frac{ds_1'}{(s_1')^{r+1}} \mu_{r,n}(s_1') = 0, \quad \forall r \geq 2n+1, \quad \forall n \geq 1, \tag{3.23}$$

where

$$\mu_{r,n}(s_1') = \sum_{k=0}^{2n} \chi_{n,k} f_{r-k}(s_1'). \tag{3.24}$$

For the 2d-Ising case, we have numerically checked that this indeed works. The story is similar for the other minimal models as shown in appendix C.

For the 2d-Ising, we observe the following for the first null constraint $n = 1, r = 3$. As mentioned above, only the $4k+1$ twists contribute in the CSDR. So the statements below are using these operators.

- The spin-0, leading twist dominates and together with the first 10 higher twists contributes $+0.335$ to the sum. The subleading twists higher spins are very tiny and can be neglected for this precision.

- The spin-2 leading twist does not contribute as its OPE coefficient is zero (this also happens for the Yang-Lee model). Spin-4 onwards contribute and all higher spins have negative sign. The sum converges to the expected answer. For $L_{max} = 30$, the contribution from non-zero spins is approximately $-0.32$ compared to the anticipated answer of $-0.335$.

### 3.5 $\phi_{pq}$ calculations

In this section, we will calculate Wilson coefficients using the CSDR eq.(3.1). This will enable us to see which operators contribute the most to a certain coefficient. We write:

$$
\begin{aligned}
F(s_1, s_2) &= \sum_{p=0, q=0}^{\infty} \phi_{pq} x^p y^q = \sum_{p=0, q=0}^{\infty} \phi_{pq}(-16ak(\tilde{z}))^p(-16a^2k(\tilde{z}))^q \\
&= \sum_{p=0, q=0}^{\infty} (-16)^{p+q} a^{p+2q} k(\tilde{z})^{p+q} \phi_{pq}.
\end{aligned} \tag{3.25}
$$

Further, using the notation of the previous section we find that

$$
F(a, \tilde{z}) = F(0, 0) + \frac{1}{2\pi i} \sum_{n=1}^{\infty} \int_{-\infty}^{-\frac{1}{4}} \frac{ds_1'}{s_1'} \beta_n\left(\frac{a}{s_1'}\right) k(\tilde{z})^n, \tag{3.26}
$$

where

$$
\beta_n\left(\frac{a}{s_1'}\right) = \sum_{k=n}^{2n} \sum_{r=k}^{2n} f_{r-k}(s_1') \chi_{n,k} \left(\frac{a}{s_1'}\right)^r, \tag{3.27}
$$

where $f$'s are defined via eq.(3.21). Here we have used the locality constraints, which is why the maximum degree of $a$ for a given power of $k(\tilde{z})$ is restricted to $2n$. Comparing we find:

$$
\phi_{pq} = \frac{(-16)^{-p-q}}{2\pi i} \sum_{r=0}^{q} \chi_{p+q, p+q+r} \int_{-\infty}^{-\frac{1}{4}} \frac{ds_1'}{(s_1')^{p+2q+1}} f_{q-r}(s_1'), \quad p+q > 0, \tag{3.28}
$$

$\phi_{00} = F(0, 0)$ which the CSDR cannot fix. Eq.(3.28) is one of the main formulas from the CSDR and we have verified that it works very nicely in a number of examples. This formula also gives the locality constraints discussed in the previous section. Namely

$$
\psi_i \equiv \phi_{pq} = 0, \ \forall p \le -1, q > |p|. \tag{3.29}
$$

We can collectively denote these locality constraints by $\psi$. So for instance $\psi_1$ will be the 1st locality constraint, $\psi_2$ the second one and so on in some chosen ordering. In appendix C, we will show how the leading one works for the minimal models.

## 4 Conformal block decomposition of Minimal models

We would like to study the s-channel conformal block decomposition for minimal models and understand its structure. With the proper normalisation, the four point function of identical operators of dimension $\frac{m-2}{2(1+m)}$ is given by

$$
G(z, \bar{z}) = ((1-z)(1-\bar{z}))^{\frac{m-1}{m+1}} \mathcal{H}\left(\frac{1}{m+1}, \frac{m}{m+1}, \frac{2}{m+1}\right) \tag{4.1}
$$

$$
- \frac{\Gamma\left(\frac{2}{m+1}\right)^2 \Gamma\left(\frac{m}{m+1}\right) \Gamma\left(\frac{2m-1}{m+1}\right) ((z-1)(\bar{z}-1))^{\frac{m-1}{m+1}} (z\bar{z})^{\frac{m-1}{m+1}}}{\Gamma\left(\frac{1}{m+1}\right) \Gamma\left(\frac{2m}{m+1}\right)^2 \Gamma\left(\frac{2-m}{m+1}\right)} \mathcal{H}\left(\frac{m}{m+1}, \frac{2-m}{m+1}, \frac{2m}{m+1}\right),
$$

with $\mathcal{H}(a, b, c) = {}_2F_1(a, b, c, z) \, {}_2F_1(a, b, c, \bar{z})$. We would like to decompose it in global two dimensional conformal blocks such as

$$
G(z, \bar{z}) = (1-z)^{\frac{m-2}{2(m+1)}} (1-\bar{z})^{\frac{m-2}{2(m+1)}} \sum_{\Delta, \ell} c_{\Delta, \ell} g_{\Delta, \ell}(z, \bar{z}), \tag{4.2}
$$

with the definition of the conformal block

$$g_{\Delta,\ell}(z,\bar{z}) = \frac{k_{\Delta+\ell}(z)k_{\Delta-\ell}(\bar{z}) + k_{\Delta-\ell}(z)k_{\Delta+\ell}(\bar{z})}{\delta_{0,\ell}+1}, \tag{4.3}$$

with $k_a(x) = x^{a/2}\,_2F_1\left(\frac{a}{2},\frac{a}{2},a,x\right)$. From the decomposition, we notice that there are two towers of operators contributing with twist $\tau = 4n$ and $\tau' = \frac{2(m-1)}{(1+m)} + 4n$ respectively, and even spin. The OPE coefficients for the first few operators with $n = 0$ are

$$c_{0,0} = 1,$$
$$c'_{0,0} = -\frac{\Gamma\left(\frac{2}{m+1}\right)^2 \Gamma\left(\frac{m}{m+1}\right)\Gamma\left(\frac{2m-1}{m+1}\right)}{\Gamma\left(\frac{1}{m+1}\right)\Gamma\left(\frac{2m}{m+1}\right)^2\Gamma\left(\frac{2-m}{m+1}\right)},$$
$$c_{0,2} = \frac{(m-2)m}{8\left(m^2+4m+3\right)},$$
$$c'_{0,2} = \frac{(m-3)m(3m-2)}{8(m+1)(3m-1)(3m+1)}c'_{0,0},$$
$$c_{0,4} = \frac{3(m-2)m^2(3m-2)}{640(m+1)^2(m+3)(3m+5)},$$
$$c'_{0,4} = \frac{m^2(m(m(m(135m-394)+11)+176)+36)}{128(m+1)^2(3m+1)(5m+1)(5m+3)(7m+3)}c'_{0,0}, \tag{4.4}$$

where we have introduced the notation $c_{n,\ell} \equiv c_{4n+\ell,\ell}$ and $c'_{n,\ell} \equiv c'_{\frac{2(m-1)}{(1+m)}+4n+\ell,\ell}$. Notice that these coefficients $c_{0,\ell}$ and $c'_{0,\ell}$ are positive for any $m > 2$. From a numerical analysis, it is possible to see that the first value of $m$ for which $c_{0,\ell}$ and $c'_{0,\ell}$ are all, except $c_{0,0}$, negative is $m = \frac{2}{3}$ corresponding to the Yang-Lee model. This is consistent with the fact that the Yang-Lee model is non-unitary.

We can use the spectrum as an input to relate the coefficient $\phi_{pq}$ of the Taylor expansion in the $x, y$ variables to the OPE data. In particular we can take

$$(1-z)^{\frac{m-2}{2(m+1)}}(1-\bar{z})^{\frac{m-2}{2(m+1)}}\left(\sum_{\ell=0}^{4}\kappa_{0,\ell}g_{\ell,\ell}(z,\bar{z}) + \sum_{\ell=0}^{4}\kappa'_{0,\ell}g_{\frac{2(m-1)}{(1+m)}+\ell,\ell}(z,\bar{z})\right) \xrightarrow{(z,\bar{z})\to(x,y)} \sum_{p,q=0}^{2}k_{pq}x^p y^q, \tag{4.5}$$

where the coefficients $\kappa_{0,\ell}$ and $\kappa'_{0,\ell}$ are arbitrary. The coefficient $k_{pq}$ are $m$-dependent linear combination of the $\kappa_{0,\ell}$ and $\kappa'_{0,\ell}$ that we would like to equate to the results from the dispersive representation to get an estimate of the OPE coefficients. For $m = 3$, we obtain for the leading two operators

$$\kappa_{0,0} = 0.105k_{00} + 6.099k_{01} + 3.516k_{10} + 3.0195k_{11} + 3.11k_{20} + 0.488k_{21}, \tag{4.6}$$
$$\kappa'_{0,0} = 1.358k_{00} - 7.961k_{01} - 4.322k_{10} - 3.9485k_{11} - 4.030k_{20} - 0.638k_{21}. \tag{4.7}$$

This allows us to make contact with the CSDR approach. When inputing the $\phi_{pq} \to k_{pq}$ we are able to get estimates for the OPE coefficients, as in (4.6). In particular, it is possible to do this procedure up to generic $\ell_{max}$, by considering greater values of $p$ and $q$.

Notice that when we expand in the $x, y$ variables there are imaginary and half-integer contributions, e.g. $iy^{1/2}$. By comparing with the expansion of the known answer, we notice that this series of terms should be absent. If we impose that these terms are negligible we can find relations between $\kappa_{0,\ell}$ and linear combinations of $\kappa'_{0,\ell}$. We will work out an example in Appendix C.

Table 1: $\phi_{pq}$ using the s-channel decomposition (4.2).

| $(m, \ell_{max})$ | $\phi_{10}$ | $\phi_{01}$ | $\phi_{11}$ | $\phi_{20}$ | $\phi_{02}$ |
|---|---|---|---|---|---|
| $(3, 0)$ | 0.335 | 0.8096 | -3.227 | -0.465 | -5.693 |
| $(3, 4)$ | 0.2499 | 0.502 | -1.640 | -0.282 | -2.675 |
| $(4, 0)$ | 0.511 | 1.141 | -4.388 | -0.628 | -7.807 |
| $(4, 4)$ | 0.365 | 0.624 | -1.889 | -0.344 | -3.020 |
| $(5, 0)$ | 0.611 | 1.291 | -4.851 | -0.688 | -8.726 |
| $(5, 4)$ | 0.423 | 0.633 | -1.821 | -0.344 | -2.894 |
| $(\frac{29}{30}, 0)$ | 0.111 | 0.370 | -1.739 | -0.240 | -3.205 |
| $(\frac{29}{30}, 4)$ | 0.1017 | 0.3408 | -1.538 | -0.212 | -2.875 |
| $(\frac{2}{3}, 0)$ | 5.437 | 19.974 | -95.587 | -12.861 | -182.895 |
| $(\frac{2}{3}, 4)$ | 5.682 | 21.271 | -105.956 | -13.744 | -206.409 |

## 4.1 $\phi_{pq}$ from s-channel OPE

By doing the OPE decomposition, we can match the conformal block decomposition with the expansion of the known correlator and then express it in $x$ and $y$. We need to include *both* towers of operators for convergence to the known answers. We obtain the table below: The last two lines represent a non unitary theory, with negative $\Delta_\sigma$. The agreement is comparable to the one of the unitary counterparts. We can make two comments. The first one is that the approximation is relatively good when we include up to spin 4, but not accurate when we only include spin 0, differently from the results that can be obtained using the dispersive integral, see table(3). Secondly, as already mentioned in addition to the $\phi_{pq}$ there are imaginary and half-integer powers in $y$ in the decomposition that we did not report in table(1). With the approximation that we are working on, such contributions are small (approximatively $10^{-3}$) but generically much larger than expected. One would expect that adding operators with higher spins could solve improve the accuracy, but actually this is not the case. In order to increase the precision, higher twists ($n = 0, 1$) needs to be included. We refer to Appendix C for an example.

If we include only one operator with twist $\frac{2m-2}{1+m}$ we get

Table 2: $\phi_{pq}$'s from s-channel representation, keeping only $\epsilon$ ($c_{0,\ell} = 0$).

| $(m, \ell_{max})$ | $\phi_{10}$ | $\phi_{01}$ | $\phi_{11}$ | $\phi_{20}$ | $\phi_{02}$ |
|---|---|---|---|---|---|
| $(3, 0)$ | 0.146 | 0.074 | -0.471 | -0.097 | -0.411 |
| $(3, 4)$ | 0.125 | 0.073 | -0.461 | -0.097 | -0.388 |
| $(4, 0)$ | 0.207 | 0.171 | -0.899 | -0.143 | -1.296 |
| $(4, 4)$ | 0.200 | 0.112 | -0.610 | -0.128 | -0.568 |
| $(5, 0)$ | 0.258 | 0.231 | -1.14 | -0.158 | -1.927 |
| $(5, 4)$ | 0.245 | 0.132 | -0.625 | -0.132 | -0.601 |

It is clear that, from the s-channel OPE, considering only one tower of operators does not give a good approximation and we need to consider both of them.

## 5 Low twist dominance in dispersive representation

Here we will tabulate for $\ell_{max} = 0$ and $\ell_{max} = 4$ the values of $\phi_{pq}$ we obtain from the dispersive integral. We retain only the leading twist $k = 0$ and **only** the $\tau = 2(m-1)/(m+1)+4k$ tower since the $\tau = 4k$ tower is projected out.

Table 3: $\phi_{pq}$'s using CSDR eq.(3.28). First line in each row is using only the $\epsilon$ operator.

| $(m,\ell_{max})$ | $\phi_{10}$ | $\phi_{01}$ | $\phi_{11}$ | $\phi_{20}$ | $\phi_{02}$ |
|---|---|---|---|---|---|
| $(3,0)$ | 0.248 | 0.504 | -1.626 | -0.281 | -2.621 |
| $(3,4)$ | 0.249 | 0.502 | -1.625 | -0.281 | -2.623 |
| $(4,0)$ | 0.332 | 0.648 | -1.847 | -0.330 | -2.843 |
| $(4,4)$ | 0.356 | 0.630 | -1.853 | -0.339 | -2.883 |
| $(5,0)$ | 0.352 | 0.676 | -1.755 | -0.321 | -2.611 |
| $(5,4)$ | 0.399 | 0.650 | -1.765 | -0.336 | -2.674 |
| $(\frac{29}{30},0)$ | 0.108 | 0.331 | -1.556 | -0.221 | -2.936 |
| $(\frac{29}{30},4)$ | 0.102 | 0.341 | -1.541 | -0.212 | -2.878 |
| $(\frac{2}{3},0)$ | 5.6708 | 21.3246 | -106.056 | -13.7349 | -206.074 |
| $(\frac{2}{3},4)$ | 5.6805 | 21.2787 | -106.025 | -13.7456 | -206.323 |

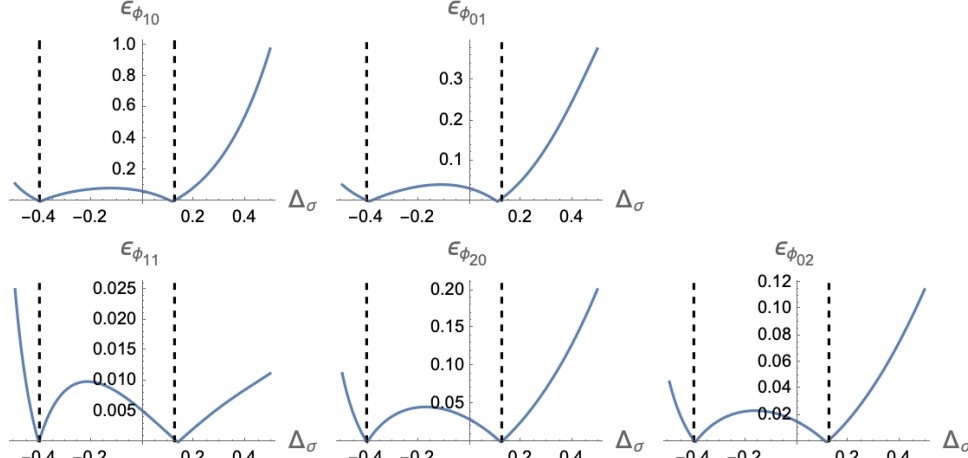

Figure 3: Fractional absolute error vs $\Delta_\sigma$. The black dashed lines indicate the Ising model and Yang-Lee model.

The last row is for the Yang-Lee model. The table demonstrates that the CSDR representation for the $\phi_{pq}$'s converges faster than the s-channel decomposition. In the QFT context, some evidence was provided in [36], although a general proof is lacking. Owing to the exponentially fast OPE convergence, low twist dominance (LTD) could have been anticipated, but we emphasise that the CSDR was crucial to demonstrate this as the s-channel convergence is not as dramatic as what is obtained from the dispersive representation. A comparison of table (3) with table (2) demonstrates this. Furthermore, to satisfy the locality constraints, we do need a large number of operators as show in appendix C. It is only for the positive Wilson coefficients $\phi_{pq}, p, q \geq 0, p + q > 0$ that LTD holds.

We can plot the absolute error defined via

$$\epsilon_{\phi_{pq}} = \left| \frac{\phi_{pq}^\epsilon - \phi_{pq}^{exact}}{\phi_{pq}^{exact}} \right|, \tag{5.1}$$

where $\phi_{pq}^\epsilon$ is obtained by putting in the dimension and OPE coefficient of the $\epsilon$ operator only in eq.(3.28). For all the $\phi_{pq}$ shown, for $\Delta_\sigma \lesssim 0.3$ the agreement is excellent, demonstrating low twist dominance (LTD). For $\phi_{10}$, LTD breaks down as $\Delta_\sigma$ approaches 0.5 but for the remaining $\phi_{pq}$ the agreement is still good. Interestingly, all plots exhibit a minimum near the Ising model

value as well as near the Yang-Lee model value. On adding more operators, the sharp feature seen below gets washed out.

# 6 Positivity, clustering and universality$_\phi$ from CSDR

We begin by noting down an amazing property of the kernel. In the dispersive integrand, we have the combination

$$\mathcal{I} = \frac{1}{s_1'}\left(\frac{s_1}{s_1'-s_1} + \frac{s_2}{s_1'-s_2}\right) = \frac{xs_1'-2y}{(s_1')^2 - xs_1' + y}, \tag{6.1}$$

where $x = s_1 + s_2, y = s_1 s_2$. We can write $a = y/x$, expand around $a = 0$ and replace $a \to y/x$ to obtain a series expansion in $x^m y^n$. We find[10]

$$\mathcal{I} = \sum_{m,n} \frac{(-1)^{m+n+1}}{(|s_1'|)^{2n+m+1}} \frac{(2n+m)\Gamma(n+m)}{n!m!} x^m y^n. \tag{6.2}$$

Now it is obvious that fixing $m + n = q$, for each $q$ we get the same sign for all the coefficients multiplying the $x, y$ powers!

To proceed, let us write the dispersive integral (in the $\tilde{z}$ variable as in eq.(3.5)) in terms of two pieces. In the first piece the integration range is from $-\infty$ to $-5/4$. In this piece we can approximate $a/s_1' \to 0$. Then we have

$$\frac{1}{2\pi i}\int_{-\infty}^{-\frac{5}{4}} \frac{ds_1'}{s_1'}\left(-16\frac{a}{s_1'}\right)\frac{\tilde{z}}{(\tilde{z}-1)^2}A(s_1',a) + \frac{1}{2\pi i}\int_{-\frac{5}{4}}^{-\frac{1}{4}} \frac{ds_1'}{s_1'}A\left(s_1',\frac{as_1'}{s_1'-a}\right)H(s_1';s_1,s_2). \tag{6.3}$$

Now notice that the first piece can only contribute to $\phi_{01}$ and $\phi_{10}$ since the kernel is now the Koebe function and in a local theory, the only terms proportional to the Koebe function are $x$ and $y$ whose coefficients are $\phi_{10}$ and $\phi_{01}$ respectively. This means that for the higher $\phi_{pq}$'s, most of the contribution will come from the second integral. Let us focus on the second integral which will control the sign of $\phi_{pq}$ for $p + q \geq 2$. It is clear that the dominant contribution will be when $A(s_1', \frac{as_1'}{s_1'-a}) \approx A(s_1', a = 0)$ since terms involving derivatives w.r.t $a$ will come with higher powers of $s_1'$ and between $-1 < s_1' < -1/4$ will be sub-leading. Next, we observe that around $s_1' = -1/4 - \epsilon$, with $\epsilon > 0$, we have for the minimal models, the lowest twist operator with twist $2\frac{m-1}{m+1}$ contributing

$$A(s_1', a = 0) = ic_{0,0}' 2^{\frac{3}{m+1}} \epsilon^{\frac{m-1}{m+1}} \sin\left(\pi\frac{m-1}{m+1}\right){}_2F_1\left(\frac{m-1}{m+1}, \frac{m-1}{m+1}, 2\frac{m-1}{m+1}, \frac{3}{4}\right) + \cdots \tag{6.4}$$

Higher twist operators will be subleading in $\epsilon$ but to conclude that they are truly subleading we will have to assume that their OPE coefficients do not overwhelm the smallness of the $\epsilon^{\#}$ factor. Some evidence for this is presented in appendix B, using the results of [34]. The sin and $_2F_1$ above are positive for $m \geq 2$. Thus the imaginary part has a *definite* sign near $s_1' \approx -1/4$. It can be further confirmed numerically that for $-5/4 \leq s_1' \leq -1/4$, the imaginary part of $A(s_1', a = 0)$ has the same sign (this is not true in the full range of the dispersive integration). Combined with the positivity property of the kernel mentioned above, we conclude that for $m + n = q \geq 2$, the $\phi_{mn}$'s have the sign $(-1)^{q+1}$. This explains the observation pointed out in section 2. For $q = 1$, we can check explicitly that the first term in eq.(6.3) does not change the conclusion. Thus we have shown that *low twist dominance* and *crossing symmetry* can explain the positivity of $(-1)^{m+n+1}\phi_{mn}$ coefficients.

---

[10]In the sum we omit the $x^0 y^0$ term as it is absent.

## 6.1 An approximate formula for $\phi_{pq}$

Using the above arguments, we can come up with an approximate formula for $\phi_{pq}$. Using eq.(6.4) and assuming that the bulk of the integral for $\phi_{pq}$ comes from the lower end of the dispersive integral, we can write[11]

$$\tilde{\phi}_{pq} \approx (-1)^{p+q+1}(2p+q)\frac{\Gamma(p+q)}{p!q!}A_0 \int_{\frac{1}{4}}^{\infty} ds_1 \frac{(s_1 - \frac{1}{4})^{\frac{m-1}{m+1}}}{s_1^{p+2q+1}}, \tag{6.5}$$

where $\tilde{\phi}$ denotes an approximation and $A_0$ is a $p,q$ independent but $m$ dependent quantity.[12] The integral can be done exactly leading to

$$\tilde{\phi}_{pq} \approx (-1)^{p+q+1}\frac{(p+2q)\Gamma(p+q)}{p!q!}4^{p+2q-\frac{m-1}{m+1}}B\left(\frac{2m}{m+1}, p+2q-\frac{m-1}{m+1}\right)A_0(m), \tag{6.6}$$

$$= (-1)^{p+q+1}\frac{(p+2q)\Gamma(p+q)}{p!q!}4^{p+2q-\frac{\Delta_\epsilon}{2}}B\left(1+\frac{\Delta_\epsilon}{2}, p+2q-\frac{\Delta_\epsilon}{2}\right)A_0(\Delta_\epsilon), \tag{6.7}$$

where $B(x,y) = \Gamma(x)\Gamma(y)/\Gamma(x+y)$ is the Euler-Beta function. This makes a prediction for the ratios of the Wilson coefficients. Defining an error

$$\epsilon_{pq} = \left| \frac{\frac{\tilde{\phi}_{pq}}{\tilde{\phi}_{p+q,0}} - \frac{\phi_{pq}}{\phi_{p+q,0}}}{\frac{\phi_{pq}}{\phi_{p+q,0}}} \right|, \tag{6.8}$$

where $\tilde{\phi}_{pq}$ is given in eq.(6.6), we obtain the plot shown in fig.(4). As is expected for lower values of $|\Delta_\sigma|$ but not for higher values. Nevertheless, considering the drastic approximations used, the formula is a reasonable approximation for a the $\phi_{pq}$'s especially for $\Delta_\sigma > 0$. The explicit formula in eq.(6.6) also enables us to check the clustering phenomena in section 2. Explicitly, notice that when we compute $\tilde{\phi}_{pq}/\tilde{\phi}_{pq,max}$ most of the $p,q$ dependence cancels out except for $\Gamma(p+2q-\frac{m-1}{m+1})/\Gamma(p+2q-\frac{\tilde{m}-1}{\tilde{m}+1})$ where $\tilde{m}$ is the value of $m$ which maximizes $\tilde{\phi}_{pq}$. This $p,q$ dependence will approximately cancel as $p,q$ become large. This explains the clustering of the plots in fig.(1).

### 6.1.1 Connecting with critical exponents

The approximate formula eq.(6.6) leads to

$$\frac{\tilde{\phi}_{10}}{\tilde{\phi}_{01}} \approx \frac{1}{2(2-\Delta_\epsilon)} = \frac{\nu}{2} = \frac{m+1}{8}, \tag{6.9}$$

where $\nu$ is a standard critical exponent. For instance, we find $\nu = 1$ for the 2d Ising model ($m = 3$) which is exactly the expected answer, while for $m = 4$ the above approximation gives $\nu = 1.26$ while the expected answer is $\nu = 1.18$. For Yang-Lee we find $0.42$ while the expected answer is $0.53$. The approximate formula tells us that for positivity to hold, even for non-unitary theories, we need $\Delta_\epsilon > -2$ which leads to

$$\nu > \frac{1}{4} \implies \frac{\tilde{\phi}_{10}}{\tilde{\phi}_{01}} > \frac{1}{8}. \tag{6.10}$$

---

[11]The replacement of the upper limit by $\infty$ yields good agreement for all $\phi_{pq}$'s except $\phi_{10}$ for large $m$ values where the integrand goes like $1/s_1^{1+\frac{2}{m}}$ and hence leads to poor convergence.

[12]Explicitly $A_0 = \frac{c'_{0,0}}{2\pi}2^{\frac{3}{m+1}}\sin\left(\pi\frac{m-1}{m+1}\right){}_2F_1\left(\frac{m-1}{m+1}, \frac{m-1}{m+1}, 2\frac{m-1}{m+1}, \frac{3}{4}\right)$.

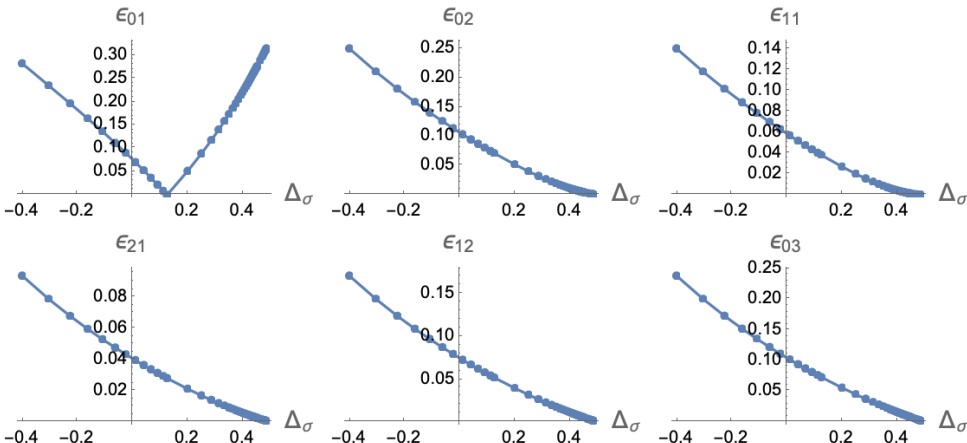

Figure 4: Comparison of eq.(6.6) with exact result.

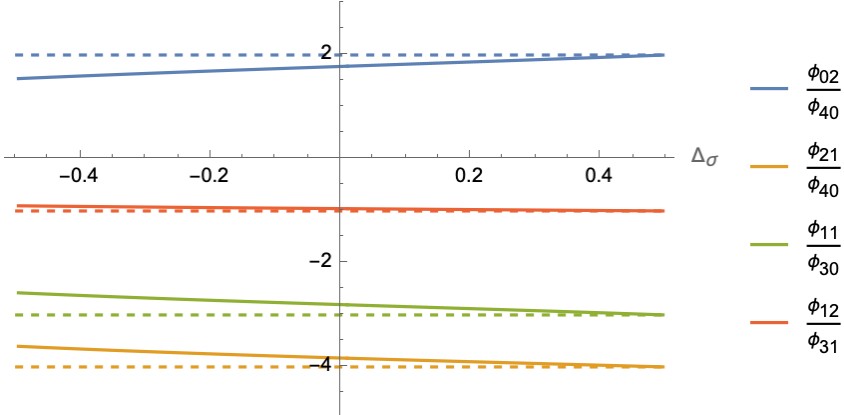

Figure 5: Universality$_\phi$ in the ratios of the Wilson coefficients. The plots are using the exact answers (solid) and match well with the LTD prediction (dashed). Fig.(4) explains the small deviations from the universality.

This bound is respected for all the 2d scenarios discussed in this paper. We also find using eq.(6.6) other approximate formulas such as

$$\frac{\tilde{\phi}_{20}}{\tilde{\phi}_{02}} \approx \frac{3}{2(\Delta_\epsilon - 4)(\Delta_\epsilon - 6)} > \frac{1}{32}, \tag{6.11}$$

where the inequality holds for $\Delta_\epsilon > -2$. For $m = 3, 4, 5, 6$ we find the values $(0.100, 0.112, 0.121, 0.128)$ while the answers from the exact expressions (table in section 2) are $(0.107, 0.118, 0.125, 0.132)$ respectively. For the Yang-Lee model, we get 0.053 while the exact answer is 0.067. As is evident the agreement in all cases is very good.

## 6.2 Universality$_\phi$ from LTD

Eq.(6.6) leads to a very interesting prediction. The $p, q$ dependence in the $m$ dependent part of the formula always appears in the combination $p + 2q$. This means that if we hold $p + 2q$ fixed then the ratios of $\phi_{pq}$'s will be universal, i.e., it will be independent of $\Delta_\sigma$ or $m$. For instance say $p + q = 4$. Then eq.(6.6) predicts that $\phi_{02}/\phi_{40} = 2$, $\phi_{12}/\phi_{31} = -1$, $\phi_{11}/\phi_{30} = -3$ and $\phi_{21}/\phi_{40} = -4$. From the plot fig.(5) with the exact known answers, we find that indeed this is respected.

## 6.3 Higher dimensions

While eq.(6.6) was derived keeping the minimal models in mind, it is easy to anticipate qualitatively what happens in higher dimensions. The power of $(s_1 - 1/4)$ in eq.(6.5) will get replaced by $\tau_m/2 - \Delta_\sigma$ as in higher dimensions we will have a different projection factor as discussed earlier. For the lower limit of the dispersive integral to be finite, we will assume $\tau_m/2 - \Delta_\sigma > -1$ which holds for the epsilon expansion. This will change the $\Delta_\epsilon$ in eq.(6.6) to $\Delta_\epsilon - 2\Delta_\sigma$ without altering the $p, q$ dependence; the $A_0(\Delta_\epsilon)$ factor will change but is not relevant for our discussion here. Thus exactly the same prediction of universality as discussed above can be anticipated when there is low twist dominance in higher dimensions. This could be used as a test for low twist dominance for any theory. In [9], for the epsilon expansion the expression for the correlator was worked out up to $O(\epsilon^2)$. Using this, one can compute the epsilon expansion for the $\phi_{pq}$'s. What is remarkable is that the $\phi_{pq}$'s respect the sign pattern[13] that is predicted by LTD *for any real $\epsilon$*! As an example we quote

$$\phi_{11} \approx 768 - 1267.56\epsilon + 907.93\epsilon^2, \quad \phi_{30} = -256 + 424.82\epsilon - 304.96\epsilon^2, \qquad (6.12)$$

using which it is easy to check that when $\epsilon$ is real $\phi_{11} > 0, \phi_{30} < 0$ and,

$$-3.01 \lesssim \frac{\phi_{11}}{\phi_{30}} \lesssim -2.98, \qquad (6.13)$$

exactly as predicted by Universality$_\phi$. The agreement for other ratios is equally impressive.[14] It will be very interesting to carry out explicit checks of this using the numerical data for 3d CFTs living on the boundary of the allowed region.

Another example that we checked is four-dimensional $\mathcal{N}$=4 Super Yang-Mills theory in the large $N$ and strong coupling regime. The sign pattern for the $\phi_{pq}$'s is completely respected in the coupling-dependent part of the correlator, and when considering only twist two, non protected operators, also the same universality of the ratios of $\phi_{pq}$'s seen for the minimal models carries over [41].

The punchline of this section is: LTD and crossing can explain the features of positivity, clustering and universality$_\phi$ pointed out in section 2.

## 7 GFT bounds for Minimal models

Here we will briefly study the Bieberbach-Rogosinski bounds [7]. The idea is to come up with a region in the $a$-parameter space where the correlator is typically real [15] inside the unit disc and hence will obey the Bieberbach-Rogosinski bounds. For QFT pion scattering, axiomatic arguments lead to the determination of this range of $a$ [6], while for massless scattering in EFTs one can determine this range numerically [8]. For CFTs however, at this point, we do not know how to make the analogous argument. As such, we will restrict our attention to studying these bounds with the known answers, rather than exploiting the bounds to constrain the theories like what was done in [6–8]. Writing

$$F = \sum_{n=0}^{\infty} \alpha_n(a) \tilde{z}^n, \qquad (7.1)$$

---

[13]Here there is an extra $(uv)^{-\Delta_\sigma}$ factor like discussed and the overall minus sign is due to the difference between 2d/4d blocks.

[14]One can reverse the logic and put bounds on the higher order terms in epsilon. For instance suppose the $O(\epsilon^2)$ term in $\phi_{11}$ was not known. Then using universality, demanding $-3.05 \leq \frac{\phi_{11}}{\phi_{30}} \leq -2.95$, we find that the $O(\epsilon^2)$ term in $\phi_{11}$ lies between 901.2 and 914.7.

[15]A typically real function is one that satisfies $Im\, f(z)\, Im\, z > 0$, see [7] for more details.

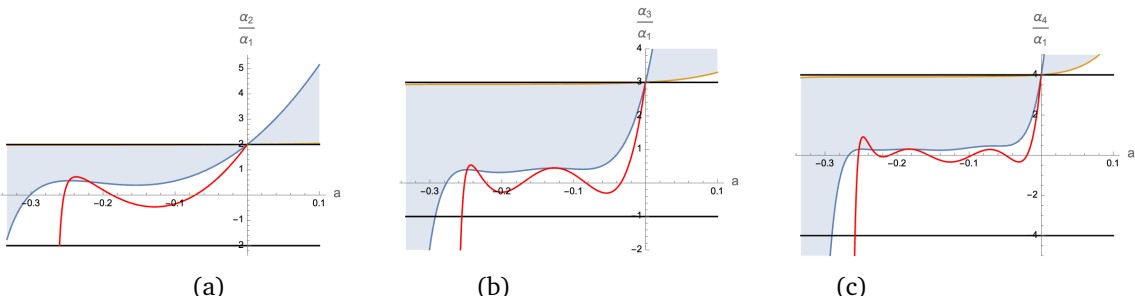

Figure 6: (a) $\frac{\alpha_2}{\alpha_1}$ vs $a$. (b) $\frac{\alpha_3}{\alpha_1}$ vs $a$. (c) $\frac{\alpha_4}{\alpha_1}$ vs $a$. The black solid lines indicate the Bieberbach-Rogosinski bounds [7]. The blue shaded regions are for unitary theories. The red lines are for the non-unitary Yang-Lee edge singularity.

the Bieberbach conjecture says that for a typically real function the $\alpha_n$'s should obey the Bieberbach-Rogosinski bounds which for $n = 2, 3$ read:

$$-2 \leq \frac{\alpha_2}{\alpha_1} \leq 2, \quad -1 \leq \frac{\alpha_3}{\alpha_1} \leq 3. \tag{7.2}$$

The plots indicate that the Minimal models populate most of the regions near the upper bound. The upper boundary is set by $\Delta_\sigma \approx 1/2$ while the lower boundary is set by $\Delta_\sigma \approx 0$. As $\Delta_\sigma \rightarrow 1/2$, $F \rightarrow 3/4 + x/2$. As a result, we get the upper bound following from the Bieberbach-Rogosinski considerations to be saturated since $x$ is proportional to the Koebe function. This is exactly what the plots indicate. As discussed above, the range $-\frac{1}{8} < a < 0$ ensures that there are no singularities inside the unit $|\tilde{z}| < 1$ disc, so in this range, we expect the GFT bounds to be respected. This is indeed verified and it is important to note that inside this range, $\alpha_p/\alpha_1$ is positive. In fact, the bounds are respected for a larger range of $a$ and this happens also for $\alpha_5/\alpha_1$ and $\alpha_6/\alpha_1$. This deserves a better explanation. The red solid lines are for the Yang-Lee non-unitary model. This indicates that the unitary models satisfy $\alpha_n$ positivity while the non-unitary ones do not.

# 8 Future directions

We conclude with possible future directions.

- In this paper we wrote down dispersion relations in the $u, v$ variables. It is also possible to consider the CFT $\rho, \bar{\rho}$ variables as in eq.(3.20) as the convergence is better in those variables.

- It will be interesting to take the diagonal limit $z \rightarrow \bar{z}$ and try to connect with the 1d work of [18, 19, 37]. In the $x, y$ variables, this corresponds to the restriction $x^2 - 2x = 4y$ and hence the Taylor expansion can be written in terms of $x$ only.

- In the future, it will be important to understand the locality constraints in more detail, both analytically and numerically. In principle, it should be possible to derive the low twist dominance from these constraints along the lines of [38]. We present more evidence for this in appendix C. In appendix C, we also show how to use the locality constraints, when we have some idea about the spectrum, but leaving the OPE coefficients undetermined. There is some evidence that the locality constraints from the CSDR are identical to the "null constraints" [4, 47] which arise on imposing crossing symmetry in the fixed-$t$ approach [5, 15]. This should continue to hold in the position space CFT case considered in the present paper.

- Extending our analysis to higher dimensions for general $\Delta_\sigma$ should be do-able, although it will need higher subtracted dispersion relations [7]. It will be interesting to see what additional restrictions lead to LTD or if the locality constraints are sufficient. We already presented evidence that the universality property will hold for the epsilon expansion at least to $O(\epsilon^2)$. It will be fascinating to probe this at the next order, presumably using the pure transcendentality ansatz used in [39]. Can universality hold at the next order? If the answer is yes, then this could be pointing at a different way to constrain the epsilon expansion order by order.

- Related to the previous point, it will be interesting to understand how to adapt this setup in the case of superconformal field theories, for instance four dimensional $\mathcal{N} = 4$ Super Yang-Mills, in the limit of large rank of the gauge group $N$ [41]. It would be interesting to see if and when the positivity is maintained and if so, how to use it as a constraint in the construction of [40], as hinted to for the epsilon expansion in the previous point.

  This program is similar in spirit to [42,43], where by inputing the conformal dimensions of the intermediate operators it is possible to compute the OPE coefficients using the conformal bootstrap and a few other constraints. It would be very interesting to see, when adapted to the suitable setup, how the results of this paper translates into explicit conditions on OPE coefficients.

- Another interesting avenue to pursue is Minimal Model Holography [44]. Here we consider coset CFTs $\frac{SU(N)_k \times SU(N)_1}{SU(N)_{k+1}}$ with $0 \leq \lambda \equiv \frac{N}{k+N} \leq 1$ in the $N, k \to \infty$ limit with central charge $c = N(1 - \lambda^2) \gg 1$. The holographic duals are higher spin gauge theories coupled to two complex scalar fields. The holographic four point functions for these complex scalar fields of conformal dimensions $\Delta_\pm = 1 \pm \lambda$ were calculated in [45]. Since there is large $N$-factorization [45], where GFF intuition kicks in, and hence the dominant exchange is the double-trace scalar with dimension $2\Delta_\pm$, it is expected that all the features discussed in this paper will carry over; a preliminary check for the $\Delta_-$ operator confirms this. A more thorough check using the techniques in [46] for non-identical operators is desirable. What will be interesting to study is what happens for moderate values of $N$, keeping $\lambda$ fixed.

- In [14], it was pointed out that in order to consider Mellin amplitudes for minimal models, one needs to subtract off an infinite number of contributions. This is analogous to the projection of the twist $4k$ operators which was in-built in our analysis. The Polyakov-Mellin bootstrap [15], is based on the measure factor in the Mellin transform leading to projecting out the GFF operators. For minimal models, our analysis suggests that a different measure factor which projects out the twist $4k$ operators should be possible. This will make the Mellin space analysis more efficient.

## Acknowledgments

We thank A. Kaviraj, M. Paulos, P. Raman and A. Zahed for discussions and A. Kaviraj and M. Paulos for useful comments on the draft. AB is supported by Knut and Alice Wallenberg Foundation under grant KAW 2016.0129 and by VR grant 2018-04438. AS acknowledges support from MHRD, Govt. of India, through a SPARC grant P315 and DST through the SERB core grant CRG/2021/000873.

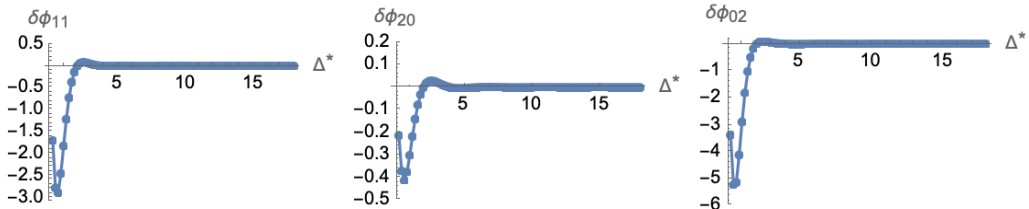

Figure 7: Estimate of $\Delta > \Delta_*$ contributions to $\phi_{pq}$. The smallest value of $\Delta_*$ we have considered is 0.2.

## A Formulas for correlators

For convenience, we give here the formula for the correlators of four $\sigma \equiv \Phi_{1,2}$'s in minimal models [29, 30]:

$$f_4 = \mathcal{N}(t) u^{\frac{t}{2}} v^{\frac{t}{2}} {}_2F_1(3t-1, t, 2t, z) {}_2F_1(3t-1, t, 2t, \bar{z})$$
$$+ u^{1-\frac{3t}{2}} v^{\frac{t}{2}} {}_2F_1(t, 1-t, 2-2t, z) {}_2F_1(t, 1-t, 2-2t, \bar{z}), \tag{A.1}$$

$$F = u^{\Delta_\sigma} v^{\Delta_\sigma} f_4, \tag{A.2}$$

where $\mathcal{N}(t) = -\frac{\Gamma^2(2-2t)\Gamma(t)\Gamma(3t-1)}{\Gamma^2(2t)\Gamma(1-t)\Gamma(2-3t)}$, and $t = p/q$ is in terms of the $p, q$ labeling the minimal model $M(q, p)$. In terms of $t$, $\Delta_\sigma = 3t/2 - 1$, $\Delta_\epsilon = 4t - 2$. The 2d Ising model is $M(4, 3)$ while the Yang-Lee model [24] is $M(5, 2)$. For Yang-Lee, $\Delta_\sigma = \Delta_\epsilon = -2/5$.

## B Estimating the contribution from large twist tails

In [34], it was shown that for unitary theories, the contribution to the four point function of identical scalar operators from operators with $\Delta > \Delta_*$ is bounded. Namely

$$|f_{\Delta > \Delta_*}| \lesssim \frac{\Delta_*^{4\Delta_\sigma}}{\Gamma(4\Delta_\sigma + 1)} \left| \frac{z}{(1+\sqrt{1-z})^2} \right|^{\Delta_*}. \tag{B.1}$$

Using this, we can find an estimate on the contribution of operators with $\Delta > \Delta_*$ on $\phi_{pq}$. The strategy is to split the contributions to the absorptive part into two pieces, $\Delta \leq \Delta_*$ and $\Delta > \Delta_*$. Then we use $Im f \leq |f|$ and the rhs of eq.(B.1), in the formulas for $\phi_{pq}$ in eq.(3.28) which arise from the *crossing symmetric dispersion relation* to estimate the contribution from the tail to $\phi_{pq}$'s. Let us write the contribution from $\Delta > \Delta_*$ by $\delta \phi_{pq}$. Then we find the following plots for the 2d Ising value $\Delta_\sigma = 1/8$:

This shows that the contribution from large twists is indeed small. In numbers, for $\Delta_* \geq 4$, we find $|\delta \phi_{11}| \lesssim 0.0012, |\delta \phi_{20}| \lesssim 0.01, |\delta \phi_{02}| \lesssim 0.008$.

Interestingly, for $\Delta_* = 1$, we find $\delta_{10} \approx 0.247, \delta_{01} \approx 0.596, \delta \phi_{11} \approx -1.82, \delta \phi_{20} \approx -0.30, \delta \phi_{02} \approx -2.884$, while the expected answers are $0.25, 0.50, -1.625, -0.281, -2.625$ respectively. Our analysis in this section may be taken as evidence for LTD. Note however, that the arguments here do not extend to non-unitary theories and hence will not explain our findings for the Yang-Lee model.

## C Yang-Lee model

The Yang-Lee model corresponds to the $M(5, 2)$ minimal model and we can study the four point function of operators of dimension $-2/5$, eg. using eq.(A.1). This correlator reduces to

the one discussed in Eq.(4.1) with $m = 2/3$, thus it can be decomposed in conformal blocks. There are two towers of operators, with twist $\tau = 4n$ and $\tau' = 4n - 2/5$, with OPE coefficients as in Eq. (4.4), provided that $m = 2/3$. This theory is non-unitary, and this is reflected by the fact that the corresponding OPE coefficients are non positive. We can read off the coefficient $\phi_{pq}$ by doing the same as in Table 1, and we get

$$\phi_{10} = 5.6825, \quad \phi_{01} = 21.2708, \quad \phi_{11} = -105.956. \tag{C.1}$$

where as the exact values are: $\phi_{10} = 5.6832, \quad \phi_{01} = 21.2624, \quad \phi_{11} = -106.017$.

Let us now consider the imaginary and half-integer terms in the $x, y$ expansion. If we include only operators with $n = 0$ and $\ell = 0, 2, 4$ in both towers, we get that for instance the term $iy$ is 0.001596, and the other first few terms are of the same order of magnitude. As already mentioned, if we insist on considering only operators with $n = 0$, we improve only mildly the convergence to zero. Instead, if we retain $n = 0, 1, 2$ with $\ell = 0, 2, 4$ we get that the first imaginary contributions is of order $10^{-6}$.

We can now use this example as an illustration of what we discuss in Section 4, namely how to use the absence of the imaginary and half-integer powers to find a relation between the OPE coefficients of the two towers of operators. Due to the excellent accuracy that our approximation has in this case (when $\ell_{max} = 0, 2, 4, 6$ and $n = 1$), we can force these unwanted terms to be strictly equal to zero. By doing so, we get a set of linear equations which can be simply solved. This gives

$$\begin{aligned} \kappa'_{0,0} &= -3.65273, & \kappa'_{0,2} &= -0.0000337402, \\ \kappa'_{0,4} &= -0.000480083, & \kappa'_{0,6} &= -0.0000198372, \end{aligned} \tag{C.2}$$

which are in excellent agreement with

$$c'_{0,0} = -3.65312, \quad c'_{0,2} = 0, \quad c'_{0,4} = -0.000492998, \quad c'_{0,6} = -0.0000160889. \tag{C.3}$$

Notice that the operators with $n$ have $c'_{1,\ell} = 0$ and we have $\kappa'_{1,\ell} \sim 10^{-7}$. This procedure can be done for arbitrarily large $n$ and $\ell_{max}$.

## D Locality constraints

Here we will examine some of the locality constraints and show that they are indeed satisfied when more operators are included. Consider the locality constraint $\phi_{-1,2} = 0$. To have evidence that it works, we will put in all leading twist ($k = 0$) operators up to some $\ell = L_{max}$. The plot gives evidence that the locality constraints will get satisfied. It is curious to note that the locality constraints become harder to satisfy beyond $\Delta_\sigma < -0.4$, which is the Yang-Lee value.

### D.1 Motivating LTD

Here we will briefly motivate the Low Twist Dominance effect starting from the locality constraint $\phi_{-1,2}$. Let us assume the tower of operators $\Delta - \ell = \Delta_\epsilon + 4k$ and focus on $k = 0$, retaining the first 8 spins for definiteness. We will let the OPE coefficients be arbitrary. Let us first examine the constraints for $m = 3$.[16] We find

$$-0.054c'_{0,0} + 19.996c'_{0,2} + 329.23c'_{0,4} + 3775.09c'_{0,6} + 42719.3c'_{0,8} + \cdots = 0. \tag{D.1}$$

---

[16]We assume here only one tower of operators in the spectrum and the dimension of the external operator for the specific value of $m$. We do not input the specific structure of the full correlator.

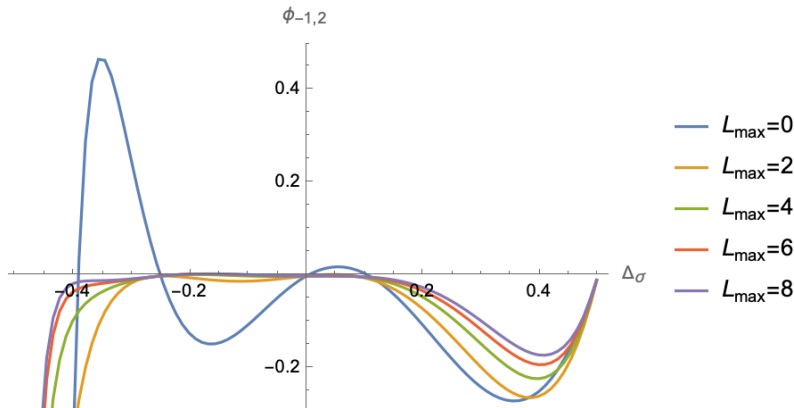

Figure 8: The locality constraint $\phi_{-1,2}$.

Thus we observe a pattern of signs whereby the higher spins all have the same sign. This automatically implies a chain of inequalities, for instance:[17]

$$-0.054c'_{0,0} + 19.996c'_{0,2} < 0 \implies \frac{c'_{0,2}}{c'_{0,0}} < 0.0027 \,, \tag{D.2}$$

$$-0.054c'_{0,0} + 329.23c'_{0,4} < 0 \implies \frac{c'_{0,4}}{c'_{0,0}} < 1.65 \times 10^{-4} \,. \tag{D.3}$$

Thus the OPE coefficients of higher spin operators are suppressed as expected, and the first locality constraint already leads to interesting bounds on the same. In the Taylor coefficients $\phi_{pq}$, $c'_{0,0}$ contributes the most. As an example we present

$$\phi_{11} = -6.503c'_{0,0} - 4.053c'_{0,2} + 31.774c'_{0,4} + 260.309c'_{0,6} + \cdots \tag{D.4}$$

From here it is clear that the contribution from $c'_{0,0}$ will be $\sim O(10^3)$ times bigger than the higher spin contributions. As another example, consider the Yang-Lee model. Here repeating the same steps as outlined above, we find[18]

$$\frac{c'_{0,2}}{c'_{0,0}} < 0.0019 \,, \quad \frac{c'_{0,4}}{c'_{0,0}} < 1.85 \times 10^{-4} \,, \tag{D.5}$$

again indicating LTD. As we increase $\Delta_\sigma$, there is a change in pattern where the spin-0, spin-2 are of one sign and the rest of the opposite sign. For instance, for $\Delta_\sigma = 0.4$, we find

$$-0.386c'_{0,0} - 0.582c'_{0,2} + 47.009c'_{0,4} + 769.70c'_{0,6} + 10358.2c'_{0,8} + \cdots = 0 \,. \tag{D.6}$$

Repeating the logic presented above, we would now keep both the spin-0 and spin-2 operators and conclude that the rest of the OPEs are suppressed compared to these two. This is perfectly consistent with our findings in fig. 3, where for increasing $\Delta_\sigma \to 0.5$, retaining only the $\epsilon$ operator led to poor agreement for $\phi_{10}, \phi_{01}$ coefficients. To promote the logic presented here to a full fledged derivation of LTD, we can retain other locality constraints in order to constrain the spectrum in addition to the OPE. We leave this for future work.

---

[17]The known answers for 2d Ising are $\frac{c'_{0,2}}{c'_{0,0}} = 0, \frac{c'_{0,4}}{c'_{0,0}} \approx 6.1 \times 10^{-5}$.

[18]The known answers for Yang-Lee are $\frac{c'_{0,2}}{c'_{0,0}} = 0, \frac{c'_{0,4}}{c'_{0,0}} \approx 1.3 \times 10^{-4}$.

## D.2   Constraining OPE using locality constraints

In most of the paper, our approach has been to come up with explanations for features observed in section 2. If we did not know the correlator to begin with, the locality constraints can be used to obtain new results with a minimal set of assumptions. As in the previous subsection, let us assume the same tower of operators with all unknown OPE coefficients. For illustrative purpose, consider $m = 3$. Now let us consider the locality constraints $\phi_{-1,2}, \phi_{-1,3}, \phi_{-1,4}, \phi_{-2,3}, \phi_{-2,4}$. Let us list these out. $\phi_{-1,2}$ was already given above.

$$
\begin{aligned}
\phi_{-1,3} = &-0.08268c'_{0,0} + 35.0115c'_{0,2} \\
&+ 745.226c'_{0,4} + 7334.46c'_{0,6} + 57313.1c'_{0,8} + \cdots = 0,
\end{aligned}
\tag{D.7}
$$

$$
\begin{aligned}
\phi_{-1,4} = &-0.40752c'_{0,0} + 128.049c'_{0,2} \\
&+ 3530.51c'_{0,4} + 40378.3c'_{0,6} + 311081.0c'_{0,8} + \cdots = 0,
\end{aligned}
\tag{D.8}
$$

$$
\begin{aligned}
\phi_{-2,3} = &\,0.2068c'_{0,0} - 65.4162c'_{0,2} \\
&- 1242.95c'_{0,4} - 14903.8c'_{0,6} - 167821.0c'_{0,8} - \cdots = 0,
\end{aligned}
\tag{D.9}
$$

$$
\begin{aligned}
\phi_{-2,4} = &\,0.41822c'_{0,0} - 148.596c'_{0,2} \\
&- 3557.23c'_{0,4} - 38870.3c'_{0,6} - 321490.0c'_{0,8} - \cdots = 0.
\end{aligned}
\tag{D.10}
$$

Notice the definite sign pattern: spin-0 is of one sign and the higher spins of opposite signs. These give

$$
0 \le \frac{c'_{0,2}}{c'_{0,0}} \le 2.4 \times 10^{-3}, \qquad 0 \le \frac{c'_{0,4}}{c'_{0,0}} \le 1.1 \times 10^{-4},
\tag{D.11}
$$

$$
0 \le \frac{c'_{0,6}}{c'_{0,0}} \le 1.0 \times 10^{-5}, \qquad 0 \le \frac{c'_{0,8}}{c'_{0,0}} \le 1.2 \times 10^{-6}.
\tag{D.12}
$$

These are expectedly stronger than what we obtained using $\phi_{-1,2}$ above. These can then be used to obtain bounds on $\phi_{pq}$'s. For instance, if we put in $c'_{0,0} = 0.25$, then we find

$$
-1.626 \le \phi_{11} \le -1.625, \quad -0.2817 \le \phi_{20} \le -0.2810,
\tag{D.13}
$$

which are in excellent agreement with the known answers. If we did not put in $c'_{0,0}$ we would get projective bounds in terms of ratios of $\phi_{pq}$'s. Thus putting in some information about the spectrum, one can derive LTD using the locality constraints.

# E   A simple application of GFT techniques

A simple application is the following. Consider expanding $F(a, \tilde{z})$ for 2d Ising around $a = 0$. We readily find

$$
F(a, \tilde{z}) = 1 - 4ak(\tilde{z}) - 8a^2\left(k(\tilde{z}) + 9k(\tilde{z})^2\right) + O(a^3),
\tag{E.1}
$$

which means that $(F(a, \tilde{z}) - 1)/(-4a)$ is just the Koebe function to leading order and will saturate the Bieberbach upper bound. Namely writing $(F(a, \tilde{z}) - 1)/(-4a) = \tilde{z} + \sum_{n=2}^{\infty} c_n \tilde{z}^n$, we will find that for $a \to 0_-$, the $c_n$'s will obey

$$
c_n \to n.
\tag{E.2}
$$

This explains the saturation of the bounds near $a \sim 0$ in fig.(6). Note that the highest power of $k(z)$ multiplying $a^n$ is $n$ and further there is a minimum power of $k(z)$ at each order in $a^n$. This is a statement of "locality"; in other words we only find positive powers of $x, y$ in the expansion.

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
