# Peer review of "Positivity, low twist dominance and CSDR for CFTs"

_SciPost Physics, doi:SciPost Phys. 14, 083 (2023)_

## Round 2 · Referee Report · Anonymous (Referee 1) · 2022-11-18

Report

In this paper the authors define in the framework of 2d diagonal unitary minimal models a function $F(u,v)$ which is symmetric under the exchange of the two variables as a consequence of the crossing symmetry of the four-point function characterized by the cross-ratios u and v. They note that the coefficients of the Taylor expansion

$$F(u,v)=\sum_{p,q} c_{p,q} x^p y^q$$
about the symmetric point $ u=v=1/4$ with $ x=u+v-1/2, y=(u-1/4)(v-1/4)$, have some surprising and unexpected properties, in particular the sign of $c_{p,q}$ is uniquely fixed by the parity of $ p+q$. These properties do not depend on unitarity, since they are also true, as the authors point out, in the 2d Yang-lee model. The authors show that these properties can be understood using a suitable dispersion relation of the mentioned symmetric function. An approximate expression of these Taylor coefficients has also been obtained. The results are interesting and most probably could be extended to a much larger class of conformal models, even with D>2, I have however some (minor) remarks: 1) F(u,v) is defined in (2.4) for the case of Isng model, but the same notation is used for more general cases. For thesake of clarity it is necessary to split equation (2.4) into two parts, the general definition and the specific form for the Ising case. 2) The interested reader could wish to know how the four-point functions have been obtained, so it is necessary to add a suitable reference for the Ising model, the minimal models reported in in eq. (4.1) and in Appendix B (mentioning the paper of Cardy where the Yang-Lee four-point function was first obtained). In conclusion, the submitted paper is interesting and also well written, so it deserves publication once the above minimal corrections are made.

Requested changes

1) Split equation (2.4) 2) Add references as specified in the report

  • validity: high
  • significance: high
  • originality: high
  • clarity: high
  • formatting: perfect
  • grammar: perfect

Author:  Aninda Sinha  on 2022-11-24  [id 3063]

(in reply to Report 1 on 2022-11-18)
Category:
remark

Thank you for the suggestions. We will incorporate them.

---

## Round 2 · Referee Report · Anonymous (Referee 2) · 2022-11-23

Report

The paper focuses on the description of certain properties of four-point functions in 2D minimal models. In particular, it considers the Taylor coefficients around the crossing-symmetric point in cross-ratio space. Starting from a position-space dispersion relation, the paper contains partial arguments that these coefficients should satisfy certain universal properties, such as sign-definiteness.

Perhaps the main interesting insight of the paper is that when the Taylor coefficients are computed using the s-channel OPE data, convergence is better when one uses the OPE inside the dispersion relation rather than directly. In fact, the coefficients are well-approximated by the contribution of a single twist tower.

However, little effort is spent on making this more precise. For example, suppose we neglect all operators of twist above some $\tau$. What is an upper bound on the error? It is stated that low-twist dominance together with the dispersion relation explains the universal features. However, this logic is somewhat circular since there is no independent proof of low-spin dominance. It would be more correct to say that the universality (observed empirically in the paper) is evidence for low spin-dominance.

I am also not sure if the quantities that the paper analyses, namely the Taylor coefficients at $u=v=1/4$, are of much independent interest. As stated in the article, the numerical conformal bootstrap does utilize such expansion, but the coefficients appearing in particular correlators (such as in minimal models) are not needed in that context.

While the paper does contain some interesting insights and computations, it does not contain ground-breaking results and it is unclear if the method/results generalize beyond the considered examples (minimal models and $\epsilon$ expansion). For these reasons, I do not recommend it for publication in SciPost.
  • validity: -
  • significance: -
  • originality: -
  • clarity: -
  • formatting: -
  • grammar: -

Author:  Aninda Sinha  on 2022-11-24  [id 3064]

(in reply to Report 2 on 2022-11-23)
Category:
remark
objection
reply to objection

  1. Regarding the error from LTD assumption, table 3 and figure 3 do precisely what the referee suggests. We have also made it clear that a derivation of LTD from the "locality" constraints is left for future work as the constraints are quite challenging. Further, we do not agree with the circular reasoning comment. By definition an argument is circular, if we use the assumptions of the argument to prove the argument. Here the assumption is LTD but we are not proving LTD. We are using LTD to explain the universal features observed in a wide range of models, both unitary and non-unitary. The fact that our observations also extend to non-unitary theories is certainly a novel element.
  2. We have already made it clear that the results of the paper are easy to generalize using similar methods. The 2d minimal model and the 4d epsilon expansion are quite different as models but both satisfy the common features observed. These features are also observed in N=4 supersymmetric 4d (which will be reported elsewhere).
  3. We were not aware the criteria for publication in Scipost is ground-breaking results (which probably are a prerequisite for PRL). The observations and explanations in this paper are novel (we are not aware of any similar work), and are a feature of at least a wide class of CFT correlators. The methods used in this paper are also novel and have not been used to study CFT correlators in position space.

Anonymous on 2022-11-25  [id 3078]

(in reply to Aninda Sinha on 2022-11-24 [id 3064])

  1. Let me clarify my suggestion. It has been shown (in https://arxiv.org/pdf/1208.6449.pdf) that when a position-space correlator is approximated by the OPE, keeping all operators up to dimension $\Delta$, the error is exponentially small in $\Delta$. My suggestion was to perform a similar analysis for the situation when the OPE is used inside the dispersion relation. I.e. what can we say about the error in general when only operators of twist up to $\tau$ are kept. In the present form, the paper observes empirically, in a particular example where the exact correlator is explicitly known, that when only the leading twist tower is kept, the error is small. However, no attempt is made at analyzing this phenomenon in general. For this reason, the claims of the paper come short of a proof. For example, in arguing for sign-definiteness of $\phi_{pq}$ after eq. (6.4), the authors say "Higher twist operators will be subleading in $\epsilon$ but to conclude that they are truly subleading we will have to assume that their OPE coefficients do not overwhelm the smallness of the factor." Have the authors considered using results on OPE convergence to justify this assumption?

  2. I disagree that the paper contains an argument whose logical structure is "low-twist dominance" $\Rightarrow$ "a nontrivial conclusion". In fact, it is not stated what the precise meaning of low-twist dominance is. Indeed, the term is not standard in the literature and as far as I can see, it is used by the authors in the sense "we get the right answer if we only keep operators of low twist". However, this is precisely what they claim the conclusion is. Indeed, the logical content of the paper is that it gives empirical evidence that low-twist operators dominate the contributions to $\phi_{pq}$. So low-twist dominance is a conclusion, not an assumption.

  3. Regarding application to other theories, I am not sure what the authors are expecting to learn. The analysis of the present paper was only possible because the four-point function of minimal models is known explicitly. What then, are we going to learn about more interesting theories where the exact four-point function is not known? This is related to my point 1 above. If it were proved on general grounds that only low-twist operators contribute significantly, one could apply this to interesting theories where the exact correlator is not known. On the other hand, if we only focus on theories where the exact correlator is known, we can simply expand it around $u=v=1/4$ and forget about the dispersion relation. So it is unclear what the benefit of the presented method is in general.

---

## Round 3 · Referee Report · Anonymous (Referee 2) · 2022-12-27

Report

The resubmitted paper has been considerably improved . The Authors
have taken into account all my remarks
and in the added appendices they have answered, at least in part, to the remarks of the other Referee. In my opinion this paper could be now published on SciPost.

---

## Round 3 · Referee Report · Anonymous (Referee 1) · 2023-1-4

Report

I appreciate the authors' effort to address my comments. I find the new version of the paper significantly better than the previous one. I am happy to change my previous suggestion and recommend that the paper is published.

---

## Round 3 · Author Response

Dear Editor,

We have incorporated the following changes in response to the referee comments.

Referee 1:

Referee 1 had accepted our paper and had some minor changes suggested. We have incorporated both. We have split eq 2.4 and have added an appendix A to address these suggestions. We have also added appropriate references.

Referee 2:

i) To clarify what we mean by LTD we have added a paragraph on pg 6. This should prevent point that the referee had confusions with.
ii) As suggested by the referee, we have looked at the tail contribution to the Taylor coefficients using the OPE convergence results. This has led to appendix B and a new plot in appendix B.
iii) The referee could not see how the present approach could lead to any new insights for situations where the correlator was not already fully known. To prevent this misimpression, we have added appendices D.1 and D.2. Here we have illustrated how the locality constraints arising from the CSDR can be gainfully used to get insights about the OPE coefficients in situations where the full correlator is not known.
iv) Considering the new analyses in points ii and iii, we have made some minor rewording throughout the text.

We do wish to emphasize that our approach was to point out a different crossing symmetric representation which seems to be better suited to address the observations in section 2 and exhibited better convergence properties than the s-channel OPE expansion. We do believe that it is a worthwhile enterprise to try to understand properties of known CFT correlators and if there are unifying explanations for such properties. In our opinion, even once you have the answer from somewhere, it is still worthwhile to understand the mathematical structure of the answer to see if there is a unifying physical picture that explains these properties (which for instance may potentially lead to a better representation). In the S-matrix bootstrap literature, there is a plethora of papers (some of which have already appeared in SciPost) which follow this philosophy, and our approach was very much in that spirit. Nevertheless, the referee’s points are well taken, and we have done our utmost best to address them.

We hope that with the new additions, the paper can be accepted for publication.

---

## Round 3 · List of Changes

Referee 1:

Referee 1 had accepted our paper and had some minor changes suggested. We have incorporated both. We have split eq 2.4 and have added an appendix A to address these suggestions. We have also added appropriate references.

Referee 2:

i) To clarify what we mean by LTD we have added a paragraph on pg 6. This should prevent point that the referee had confusions with.
ii) As suggested by the referee, we have looked at the tail contribution to the Taylor coefficients using the OPE convergence results. This has led to appendix B and a new plot in appendix B.
iii) The referee could not see how the present approach could lead to any new insights for situations where the correlator was not already fully known. To prevent this misimpression, we have added appendices D.1 and D.2. Here we have illustrated how the locality constraints arising from the CSDR can be gainfully used to get insights about the OPE coefficients in situations where the full correlator is not known.
iv) Considering the new analyses in points ii and iii, we have made some minor rewording throughout the text.

---

## Editorial Decision

published